

# Online molecular characterisation of organic aerosols in an atmospheric chamber using Extractive Electrospray Ionisation Mass Spectrometry

Peter J. Gallimore[1], Chiara Giorio[1,2], Brendan M. Mahon[1] and Markus Kalberer[1]

[1] Department of Chemistry, University of Cambridge, Lensfield Road, Cambridge, CB2 1EW, UK.
[2] now at Aix Marseille Univ, CNRS, LCE Marseille, France.

*Correspondence to*: Markus Kalberer (markus.kalberer@atm.ch.cam.ac.uk)

**Abstract.** The oxidation of biogenic volatile organic compounds (VOCs) represents a substantial
source of secondary organic aerosol (SOA) in the atmosphere. In this study, we present online measurements of the molecular constituents formed in the gas and aerosol phases during α-pinene oxidation in the Cambridge Atmospheric Simulation Chamber (CASC). We focus on characterising the performance of Extractive Electrospray Ionisation (EESI) mass spectrometry (MS) for particle analysis. A number of new aspects of EESI-MS performance are considered here. We show that
relative quantification of organic analytes can be achieved in mixed organic-inorganic particles. A comprehensive assignment of mass spectra for α-pinene derived SOA in both positive and negative ion modes is obtained using an ultra-high resolution mass spectrometer. We compare these online spectra to conventional offline ESI-MS and find good agreement in terms of the compounds identified, without the need for complex sample work-up procedures. High time resolution (7
minutes) EESI-MS spectra are compared with simulations from the near explicit Master Chemical Mechanism (MCM) for a range of reaction conditions. We show that MS intensities scale with modelled concentrations for condensable products (pinonic acid, pinic acid, OH-pinonic acid). Relative quantification is achieved throughout SOA formation as the composition, size and mass (5-2400 μg m$^{-3}$) of particles is evolving. This work provides a robust demonstration of the advantages
of EESI-MS for chamber studies over offline ESI-MS (time resolution, relative quantification) and over "hard" online techniques (molecular information).

## 1 Introduction

Airborne particulate matter has significant impacts on global climate (Hallquist et al., 2009) and human health (Dominici et al., 2006). Organic compounds typically comprise around 50% of



submicron aerosol mass (Jimenez et al., 2009). Most of this is secondary and biogenic in origin (Hallquist et al., 2009); the oxidation of biogenic volatile organic compounds (VOCs) such as monoterpenes and isoprene represents a major source of atmospheric secondary organic aerosol (SOA) (Kroll and Seinfeld, 2008; Ziemann and Atkinson, 2012). However, SOA formation

processes remain highly uncertain and this is regarded as a major weakness in the current understanding and model representation of atmospheric aerosols (Boucher et al., 2013). The chemistry involved is complex, and the range of organic compounds present in the atmosphere is extremely diverse (Goldstein and Galbally, 2007). Understanding how SOA components form and react is therefore a conceptual and analytical challenge (Noziere et al., 2015).

Identifying and quantifying individual organic components in this complex mixture is commonly achieved using mass spectrometry (MS). Of particular utility for understanding organic reaction mechanisms are so-called "soft" ionisation techniques, which retain molecular structure during ion formation (Zahardis et al., 2011). Most conventional soft ionisation MS is "offline", where chemical

analysis is performed subsequent to sampling. Techniques such as electrospray ionisation (ESI) MS have yielded extensive insight into aerosol chemical processes (Claeys et al., 2009; Edney et al., 2005; Kampf et al., 2012; Kourtchev et al., 2014). However, there are drawbacks to offline ESI-MS: limited time resolution of typically an hour or more, the potential for sampling artefacts (Turpin et al., 2000), and a lack of analyte quantification when used in a direct-infusion mode.

A number of recent studies have therefore developed online or semi-continuous atmospheric-pressure ionisation (API) MS techniques (Bateman et al., 2010; Brüggemann et al., 2015; Clark et al., 2014; Pereira et al., 2014; Vogel et al., 2013; Zhang et al., 2015). In this paper, we focus on the application of one such approach to organic aerosol analysis, namely extractive electrospray

ionisation (EESI) MS. The EESI process predominantly forms molecular ions ([M + H]$^+$ and [M − H]$^-$) and is able to efficiently ionise organic analytes even in complex sample matrices such as urine (Devenport et al., 2014), beer (Zhu et al., 2010) and olive oil (Law et al., 2010). Doezema et al., (2012) demonstrated an initial application of EESI-MS to organic aerosols, identifying a number of products formed from α-pinene ozonolysis in positive ion mode. Horan et al., (2012) reported a

related technique for particle and gas phase chemical characterisation, ambient electrospray ionisation (AESI).

Our earlier study reported the first quantification of the EESI aerosol extraction process for carboxylic acid particles (Gallimore and Kalberer, 2013). The detected MS intensity scaled in direct

proportion with the aerosol mass concentration and was independent of particle diameter in the



ranges studied (3-600 µg m$^{-3}$, 70-200 nm). Gallimore et al., (2017) applied the technique for the first time to characterising the kinetics of particle-phase reactions, using the ozonolysis of oleic acid particles as a model system. Changes in relative intensities of ions were used successfully as a proxy for relative concentrations and complex oligomeric species could be detected with minimal
molecular fragmentation. Very recently, we demonstrated the ability of the ion source to stably operate for several hours during the evolution of limonene SOA (Gallimore et al., 2017b).

In the current study, we evaluate in detail the use of EESI-MS for atmospheric chamber experiments. Large-volume atmospheric chambers have proven to be a valuable means of exploring volatile
organic compound (VOC) oxidation mechanisms because a simplified subset of reactions can be investigated under well-defined conditions (Cocker et al., 2001; Gallimore et al., 2017b; Paulsen et al., 2005). We investigate SOA formation from the ozone-initiated oxidation of α-pinene, the most abundant monoterpene in the troposphere (with estimated emissions ~60 Tg yr$^{-1}$, Guenther et al., (2012)). α-pinene forms SOA from the condensation of low volatility oxidised products, and is a
major source of biogenic aerosol (Claeys et al., 2009; Jenkin, 2004; Kristensen et al., 2013). Species identified here are compared to previous experimental studies and model representations of the oxidation pathways.

Ultra-high resolution mass spectra of α-pinene SOA are obtained using EESI operating in both ion
polarities (+ and –). EESI-MS is used alongside Proton Transfer Reaction (PTR) MS (Blake et al., 2009) to obtain molecular-level gas and particle phase information with high time resolution. We are particularly interested in the ability of EESI-MS to provide quantitative information such as concentration changes during aerosol formation and aging. We initially confirm that relative quantification of individual organic species is still possible in mixed organic-inorganic particles.
This is then extended to chemically complex chamber SOA, where the particle composition, size and mass is evolving over time. In particular, we find that EESI(–) MS intensities of condensable products of α-pinene oxidation (pinonic acid, pinic acid, OH-pinonic acid) map quantitatively onto simulated concentrations from the Master Chemical Mechanism (Jenkin, 2004). These findings support the continued use of EESI-MS for chamber experiments and prompt further development of
the technique to improve sensitivity for other applications.





## 2 Methods

### 2.1 Quantification of mixed organic-inorganic particles with EESI

An aerosol generation system, described in detail in Gallimore and Kalberer (2013) was used to produce model aerosols for quantifying the extraction and ionisation of organic compounds in the presence of inorganic salts (Figure S1).

Aerosols were produced from aqueous solutions using a custom made constant-output atomiser. Solutions containing L-tartaric acid (99 %, Aldrich) and ammonium sulfate (99.5 %, Fluka) in water (HPLC grade, Rathburn) were prepared. The total solute concentration was held constant at 0.1 mol $L^{-1}$ and four stock solutions were prepared with the following tartaric acid mole fractions ($x_{TA}$): 1, 0.98, 0.9, 0.5, the remainder being composed of ammonium sulfate ($x_{AS} = 1 - x_{TA}$). The nebuliser was supplied with $N_2$ (oxygen-free nitrogen, BOC) at a pressure of 3 bar to produce an output flow rate of 1.3 L $min^{-1}$.

A silica diffusion dryer was used to produce dry particles (<10 % RH). The dried polydisperse particles were size-selected in the range 50-200 nm prior to EESI-MS analysis using a differential mobility analyser (DMA) (TSI model 3081). The outflow from the DMA was split, with 0.3 L/min sampled by a condensation particle counter (CPC) (TSI model 3775) to measure the size-selected particle concentration. The remaining 1 L $min^{-1}$ was introduced into the EESI source. Particle mass concentrations were calculated from the CPC number concentration by assuming an aerosol density of 1.78 g $cm^{-3}$, close to the bulk densities of tartaric acid (1.79 g $cm^{-3}$) and ammonium sulfate (1.77 g $cm^{-3}$). Particles were assumed to be internally mixed with a composition representative of the bulk nebuliser solution. The mass concentrations were corrected for the transmission of multiply-charged particles through the DMA according to the method described in Gallimore and Kalberer (2013).

### 2.2 Atmospheric chamber operation

Experiments on the oxidation of biogenic VOCs were performed in the newly commissioned Cambridge Atmospheric Simulation Chamber (CASC) which is characterised in detail in Gallimore et al., (2017b). Aspects of the chamber operation relevant to the results in this paper are described briefly here (Figure S2).

The chamber consists of a 5.4 $m^3$ Fluorinated Ethylene Propylene (FEP) bag housed in an aluminium frame. Gas introduction and sampling are achieved through stainless steel flanges containing



Swagelok fittings at the front and rear of the chamber. A series of mass flow controllers (MKS, 5-200 L min$^{-1}$) are used to control flows into the chamber. The frame houses opaque perspex screens to block light from entering the chamber during "dark" reactions. The chamber is cleaned between experiments by flushing clean air from a zero air generator (Parker Hannifin KA-MT2) through the

chamber using a mass flow controller and pump (Charles Austen ET200) at matched 200 L min$^{-1}$ flows. Ozone and "hard" UV lights may also optionally be used during cleaning.

Aerosol formation was investigated for the dark reaction between α-pinene and ozone in the presence of inorganic seed particles. The chamber was filled with clean air from the zero air

generator. Water vapour was introduced by bubbling air at 5 L min$^{-1}$ through a heated round-bottomed flask containing water. The chamber's relative humidity (RH) and temperature were monitored using a Honeywell HIH4000 probe and were typically 60% RH and 291 K respectively in these experiments. Seed aerosols were produced from ammonium sulfate solutions using a nebuliser and dryer as described above. The seed particle concentration in the chamber was ~5 μg

m$^{-3}$. α-pinene (98 %, Aldrich) was evaporated into the chamber from an impinger using clean air. Ozone was produced by flowing air through a photolysis tube containing a mercury UV lamp (Appleton Woods) at 10 L min$^{-1}$, corresponding to a change in the chamber [O$_3$] of ~50 ppb min$^{-1}$. The concentration in the chamber was monitored using a photometric ozone analyser (Thermo Scientific model 49i) and was introduced until [O$_3$]/[α-pinene]$_0$ ~ 3 was achieved (~3-30 minutes).

The range of α-pinene and ozone concentrations used in the chamber are detailed in Table 1.

| Experiment | [α-pinene]$_0$ (ppb) | [O$_3$]$_{max}$ (ppb) | Online MS sampling |
|---|---|---|---|
| Low | 45 | 145 | PTR, EESI(–) |
| Medium 1 | 99 | 320 | PTR, EESI(–) |
| Medium 2 | 100* | 325 | EESI(+) |
| High | 502 | 1450 | PTR, EESI(–) |

**Table 1: Conditions used in the chamber experiments in this paper. In all cases, the chamber humidity was adjusted to 60% RH and ~ 5 μg m$^{-3}$ ammonium sulfate seed particles were introduced prior to α-pinene and ozone introduction. (*) Estimated concentration since PTR-MS not used in this experiment.**

An air "sprinkler" system, consisting of a 2 m PTFE tube with a series of small holes along its length, was supplied with high pressure bursts of air to mix the chamber constituents without recourse to a fan. The air sprinkler leads to the addition of ~100 L clean air during ozone introduction, and results in a mixing time of a few minutes (Gallimore et al., 2017b). The size distribution of aerosols in the

chamber was monitored using an SMPS (TSI model 3936). To enable presentation of data as





measured, including "raw" EESI-MS time series, wall-loss correction was not attempted and therefore we focus primarily on the initial time following ozone introduction, where aerosol production will dominate over loss to the chamber walls.

**2.3 EESI source operation and chamber sampling**

The EESI source is described in detail in Gallimore and Kalberer (2013). Briefly, it consists of a custom-built aerosol injector and housing which is interfaced with a commercially-available electrospray ionisation (ESI) source. The primary solvent electrospray can generate droplets with positive or negative charges depending on the potential difference between the ESI probe and the
mass spectrometer. Here the primary solvent was a water-methanol 1:1 mixture (Optima LC grade solvents, Fisher Scientific) containing 0.05 % formic acid (90 %, Breckland) as a spray modifier. The nitrogen sheath gas flow rate was set to 0.8 L min$^{-1}$. The potential difference was set to +3.0 and –3.0 kV for positive and negative ion modes respectively. We refer to these operating conditions as EESI(+) and EESI(–) for the remainder of the manuscript.

The aerosol injector delivers particles into the primary solvent spray at a flow rate of 1 L min$^{-1}$. Particle-droplet collisions dissolve the aerosol analytes, which are ionised and ejected into the gas phase by a Coulomb explosion mechanism. The commercial ESI housing was found not to be air tight, so a batch sampling procedure was adopted to introduce particles from the chamber into the
EESI source (Figure 1).

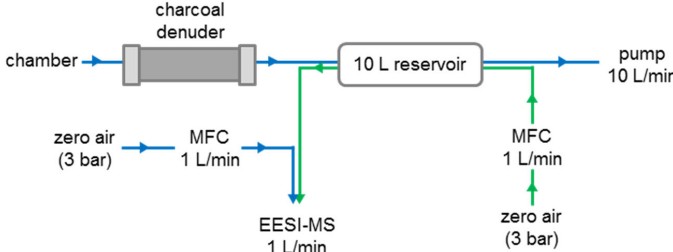

**Figure 1: Batch sampling system used to supply the EESI source with chamber air in a two-step process. 1. (blue lines): air is drawn from the chamber into an intermediate 10 L reservoir, during which time the**
**EESI source is flushed with zero air and a blank spectrum is acquired (3.5 minutes). 2. (green lines): chamber air from the flow tube is pushed through to the EESI source to acquire a sample spectrum (3.5 minutes).**





Air was drawn from the chamber at 10 L min⁻¹ through a charcoal denuder, used to remove ozone and VOCs, into an intermediate reservoir of approximately 10 L. During this time, the aerosol injector and EESI source were flushed with 1 L min⁻¹ synthetic air (zero-grade, BOC) to obtain a "blank" mass spectrum and maintain a constant gas flow into the source (blue flow configuration in

Figure 1). After 3.5 minutes, the air flush and pump were switched off and three-way valves were used to isolate the reservoir from the chamber and connect it to the EESI source. Air was then pushed through the flow tube and into the source at 1 L min⁻¹ and a sample mass spectrum acquired (green flow configuration in Figure 1).

Repetition of this cycle allowed batch sampling with a time resolution of 7 minutes. Particle losses using the sampling system in this way were characterised using an SMPS and were ~10 % of the total aerosol mass. Chamber air in the reservoir volume was not expected to be diluted significantly by the 1 L min⁻¹ inflow during sample acquisition under the laminar flow conditions used (discussed in the Supplementary Information).

### 2.4 SOA filter sampling and analysis

Filter samples of SOA were collected during the same experiments as the online composition measurements. The sampling and analysis protocol was based on that described in Kourtchev et al., (2014). Briefly, particles were drawn through a charcoal denuder and collected onto cleaned quartz

fibre filters (Pallflex® Tissuquartz 2500QAT-UP, 47 mm diameter) 1 hour after the introduction of ozone to the chamber. Chamber air was collected for 30 minutes at 15 L min⁻¹, resulting in a total volume collected of 450 L. One quarter of the filter was extracted in 2 mL methanol (LC-MS grade, Fisher Scientific) using a vortex mixer. 2 mL water (LC-MS grade, Fisher Scientific) was added to the extract, which was analysed by direct infusion nano-ESI (Advion Triversa Nanomate) MS with

a back pressure of 0.8 psi and an ionisation potential difference of –1.4 kV.

### 2.5 Ultra-high resolution MS operation and data analysis

The EESI and nanoESI sources were coupled to an ultra-high resolution mass spectrometer (Thermo Scientific LTQ Orbitrap Velos). Mass spectra were acquired in the range $m/z$ 100-500 with mass

accuracy <1.5 ppm and resolution of 100 000 (full width at half maximum, FWHM) at $m/z$ 400. These characteristics mean that unambiguous molecular formula assignments for reaction product ions can be routinely achieved.



Mass spectra generated from EESI and offline nanoESI samples were analysed using a method similar to that described in Zielinski et al., (2017). Briefly, possible formulae were assigned to the spectra using XCalibur 2.1 software (Thermo Scientific). Evaluation of these initial assignments

was performed using an in-house code run in Mathematica 10 (Wolfram Research Inc.) which removed formulae deemed implausible based on their atomic ratios. Following this assignment, the blank spectrum was subtracted and low intensity peaks deemed to fall below the noise of the Orbitrap instrument were removed to yield the final spectra.

Time series of individual particle-phase ions were also extracted from Xcalibur. These raw time series were processed by removing the transitions between sample and blank periods (~0.5 minutes), averaging the intensity of the remaining sample period, and subtracting the averaged blank intensity for the minute preceding each sample. The resulting data points, with 7 minute time resolution, were shifted in time to account for the delay between sampling from the chamber and detection by EESI-

MS. Uncertainties on the MS intensities represent the combined standard deviation of the sample and blank signals.

### 2.6 PTR-MS operation and data analysis

Gas phase VOC concentrations were measured using a proton transfer reaction mass spectrometer

(PTR-ToF-MS 8000, Ionicon Analytik, Innsbruck, Austria) in the range $m/z$ 10-500 and with a time resolution of 10 s. Parameters of the PTR-MS during the experiments were as follows: drift tube voltage: 600 V, drift tube pressure ≈ 2.20 mbar, drift tube temperature: 60°C, resulting in an E/N of ca. 135 Td (1 Td = $10^{-17}$ V cm$^2$). Resolution in the Time-of-Flight (ToF) detector was 5000 (FWHM) at the mass of protonated acetone during all experiments. The MS inlet (PEEK tube, 60 °C, flow rate

0.1 L min$^{-1}$) was connected to the smog chamber with a 1mm inner diameter PTFE tube at room temperature.

Data analysis for the PTR-MS was carried out using PTR-MS Viewer 3.2 (Ionicon Analytik). Mass calibration has been adjusted using $H_3^{18}O^+$ ($m/z$ = 21.023), $NO^+$ ($m/z$ = 29.998) and $C_3H_7O^+$ ($m/z$ =

59.049) as references. For all the compounds, concentrations were estimated on the basis of the rate constant ($k$) of the proton transfer reaction (Lindinger et al., 1998) which is essentially limited by gas-phase diffusion. Because the proton transfer reaction rate constants are not known for all



compounds, a default rate constant ($k$) of $2\times10^{-9}$ cm$^3$ molecule$^{-1}$ s$^{-1}$ was used for those compounds without a measured rate constant.

### 2.7 Numerical modelling of chamber chemistry

The complete reaction scheme for the degradation of α-pinene was extracted from the Master Chemical Mechanism (MCM) v3.3.1 (Jenkin et al., 1997; Saunders et al., 2003) via the MCM website (http://mcm.leeds.ac.uk/MCM) and used to simulate the ozone-initiated oxidation of α-pinene in the chamber. The reaction scheme was modified to approximate the introduction of ozone over the initial minutes of our experiments – an ozone precursor with initial concentration [O$_3$]$_{max}$

(from Table 1) was added to the mechanism and converted to ozone on the appropriate timescale (3-30 minutes). Simulations were performed using the box model AtChem (https://atchem.leeds.ac.uk) via a web interface which enables the use of MCM mechanisms and relevant input parameters from the chamber. AtChem uses the Fortran CVODE library to integrate the MCM reaction scheme ODEs forward in time from the initial input conditions.

AtChem simulates gas-phase chemistry, but not aerosol formation. To compare gas-phase concentrations from the simulation with EESI-MS aerosol measurements, we neglected possible in-particle chemistry and focused on major aerosol components from previous studies. Gas-particle partitioning considerations are discussed in section 3.3.2. The AtChem output concentrations

(molecules cm$^{-3}$) were converted to parts per billion by volume (ppb) for comparison to PTR-MS measurements and μg m$^{-3}$ for aerosol species.

### 3. Results

### 3.1 Relative quantification of organic analytes in mixed organic-inorganic particles

EESI-MS approaches have demonstrated excellent tolerance to very complex sample matrices compared to direct ESI-MS (Chen et al., 2006). Here we investigate the possible impact of inorganic salts on the EESI-MS intensities of organic ions in mixed aerosol particles. Measurements quantifying the response of the technique to particles containing known mixtures of tartaric acid and ammonium sulfate are given in Figure 2. Tartaric acid was detected as an [M – H]$^-$ ion in negative

ion mode at $m/z$ 149.0092. The intensity of this ion is plotted as a function of tartaric acid aerosol



mass concentration, calculated from the total aerosol mass concentration and the tartaric acid mass fraction, in Figure 2.

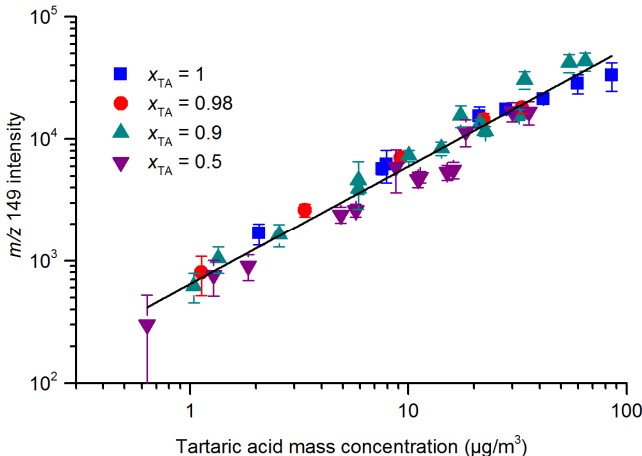

**Figure 2: Intensity of *m/z* 149.0092 in negative ion mode, assigned to deprotonated tartaric acid, as a function of tartaric acid mass concentration in the aerosol. Particles were produced by nebulising bulk solutions containing varying proportions of tartaric acid and ammonium sulfate. The mole fraction of tartaric acid was varied between 0.5 and 1. Error bars represent the standard deviation of the MS signal over the 1 minute averaging time.**

As for the single-component aerosol in Gallimore and Kalberer (2013), the detected mass spectrum signal intensity scales linearly with the organic aerosol mass concentration, this time over a range of organic/inorganic aerosol fractions. The best fit curve for the entire dataset in Figure 2 follows a power law with an exponent of 0.97, close to the value of 1 expected for such a linear relationship.

The data in Figure 2 are colour coded according to the mole fraction of tartaric acid present, $x_{TA}$. To a good approximation, the data for different $x_{TA}$, and hence different $x_{AS,}$ cluster around a common best fit curve suggesting that the different mass fractions of ammonium sulfate do not influence the ionisation efficiency (IE) of tartaric acid. There is limited evidence that higher concentrations of ammonium sulfate may suppress the organic signal; the data with $x_{TA} = x_{AS} = 0.5$ often fall slightly

below the best fit curve. Figure S3 suggests that this suppression may result in a signal up to 25% lower than expected at the highest ammonium sulfate concentrations used.





This is not problematic for the current application because the conditions used in the chamber experiments when EESI-MS measurements were made involved low ammonium sulfate concentrations (~5 µg m$^{-3}$) and high organic mole fractions ($x_{org}$ > 0.7). Although care may be required if using substantially higher salt concentrations (or higher $x_{salt}$) in future studies, the

potential suppression effect seems rather limited over the range tested. In this study, we further investigate the relative quantification of individual organic species during the growth of α-pinene SOA on ammonium sulfate seeds in section 3.3.2.

**3.2 Molecular characterisation of α-pinene oxidation products**

**3.2.1 Assignment and comparison of EESI-MS and PTR-MS spectra**

EESI-MS (particle phase) and PTR-MS (gas phase) were deployed during dark α-pinene ozonolysis experiments in the CASC chamber. Our EESI source was operated in both positive (+) and negative (–) ionisation modes for the first time which can help to identify complementary compound classes. A summary of the most abundant species identified during the "medium" concentration experiments

(100 ppb α-pinene) is presented in Table 2. Full mass spectra are also shown in Figure S4 for reference. In total, ~400 ions were detected via PTR-MS including product parent ions, fragments and contaminants. >1000 ions were detected in both EESI-MS polarities, with 95 assignments retained in EESI(+), and 87 in EESI(–), following data treatment.

| Neutral Mass (Da) | Formula | Possible assignment | PTR-MS ions | EESI-MS ions |
|---|---|---|---|---|
| 136.1252 | $C_{10}H_{16}$ | α-pinene[a,b] | $[M+H]^+$; $[C_6H_8+H]^+$ | ND |
| 140.0837 | $C_8H_{12}O_2$ | 2,2-dimethyl-cyclobutyl-1,3diethanal[c] | $[M+H]^+$ | $[M+H]^+$; $[M+Na]^+$; $[M–H]^-$ |
| 140.1201 | $C_9H_{16}O$ | 1-acetyl-2,2,3-trimethylcyclobutane[c] | $[M+H]^+$ | $[M+H]^+$; $[M–H]^-$ |
| 154.0994 | $C_9H_{14}O_2$ | Norpinaldehyde[c,f] | $[M+H]^+$ | $[M+H]^+$; $[M+Na]^+$; $[M–H]^-$ |



| 158.0943 | $C_8H_{14}O_3$ | 2,2-dimethyl-3-formyl-acyclobutylmethanoic acid[c] | ND | [M+Na]$^+$; [M–H]$^-$ |
|---|---|---|---|---|
| 168.1150 | $C_{10}H_{16}O_2$ | Pinonaldehyde[a,b,c,e,f] | [M+H]$^+$; [M–H$_2$O+H]$^+$; [C$_8$H$_{10}$+H]$^+$ | [M+H]$^+$; [M+Na]$^+$; [M–H$_2$O+H]$^+$ |
| 170.0943 | $C_9H_{14}O_3$ | Pinalic-3-acid[e,f] | [M+H]$^+$ | [M+Na]$^+$; [M–H]$^-$ |
| 172.0743 | $C_8H_{12}O_4$ | Terpenylic acid[c,d,g] | [M+H]$^+$ | [M+H]$^+$; [M+Na]$^+$; [M–H]$^-$ |
| 182.0943 | $C_{10}H_{14}O_3$ | Oxopinonaldehyde[c] | [M+H]$^+$ | [M+H]$^+$; [M+Na]$^+$; [M–H]$^-$ |
| 184.1099 | $C_{10}H_{16}O_3$ | Pinonic acid[c,d,e,f] | [M+H]$^+$ | [M+H]$^+$; [M+Na]$^+$ [M–H]$^-$ |
| 186.0892 | $C_9H_{14}O_4$ | Pinic acid[c,d,e,f] | [M+H]$^+$ | [M+H]$^+$; [M+Na]$^+$ [M–H]$^-$ |
| 188.0685 | $C_8H_{12}O_5$ | 2-hydroxyterpenylic acid[c,g] | ND | [M–H]$^-$ |
| 200.1049 | $C_{10}H_{16}O_4$ | OH-pinonic acid[c,d,e,f] | [M+H]$^+$ | [M+Na]$^+$; [M–H]$^-$ |
| 232.0947 | $C_{10}H_{16}O_6$ | Diaterpenylic acid acetate[c,d,g] | ND | [M+Na]$^+$; [M–H]$^-$ |
| 358.1628 | $C_{17}H_{26}O_8$ | Pinyl-diterpenylic ester[d] | ND | [M–H]$^-$ |

**Table 2: Tentative assignments of a selection of major ions detected by PTR-MS and EESI-MS during dark α-pinene ozonolysis experiments with 100 ppb α-pinene precursor. ND = not detected. Compounds assigned in other studies were used to provide possible assignments here: a – Wisthaler et al., (2001); b – Lee et al., (2006); c – Hall IV and Johnston (2012); d – Kristensen et al., (2013), e – Jenkin (2004), f –**
**Camredon et al., (2010), g – Claeys et al., (2009).**

The major products identified by EESI-MS and PTR-MS following data treatment compare well to previous literature. Assigning PTR-MS spectra is slightly complicated by fragmentation; abundant products such as pinonaldehyde appear mostly as fragment ions (Wisthaler et al., 2001). Since

fragmentation patterns for most VOCs are not known, we have not assigned ions < *m/z* 100 aside from known major species reported in Blake et al., (2009). However, we present all detected PTR-MS ions in Figure S4.

A positive characteristic of EESI-MS is that most species are detected as intact quasi-molecular ions

(Table 2). Furthermore, the two ion polarities allow detection of complementary compound classes. EESI(+) mostly forms H$^+$ and Na$^+$ clusters with the parent molecule and enables a wide range of



functional groups (carbonyls, alcohols, carboxylic acids) to be detected (Table 2). Doezema et al., (2012) reported that $[M-H_2O+H]^+$ fragment ions of major products such as pinonic acid and pinonaldehyde were more abundant than corresponding $[M+H]^+$ peaks. We found that such fragments were minor compared to quasi-molecular ions, and not observed for many species. This

suggests that the choice of EESI conditions, such as a lower spray voltage in the current study, may be an important determinant of fragmentation. Negative ionisation is a more specific technique which mostly deprotonates acidic functional groups and so carboxylic acids (including multifunctional species) are readily detected as $[M-H]^-$ ions.

Taken together, EESI(–)-MS, EESI(+)-MS and PTR-MS enable the detection of a wide range of typical SOA components and gas-phase oxidation products with different volatilities and polarities. We illustrate this using the two dimensional formulation introduced by Kroll et al., (2011). This calculates a compound's average carbon oxidation state, $\overline{OS_c}$, as a metric for its degree of oxidation. For molecules containing only carbon, hydrogen and oxygen:

$$\overline{OS_c} = \frac{2n_O}{n_C} - \frac{n_H}{n_C} \qquad (1)$$

Where $n_O$, $n_C$ and $n_H$ are the number of oxygen, carbon and hydrogen atoms respectively. This expression is exact unless oxygen is present in peroxide or other functional groups with oxidation state $\neq 2$. Figure 3 shows calculated $\overline{OS_c}$ from PTR, EESI(+) and EESI(–) MS assignments as a function of $n_C$. Many PTR-MS ions from Figure S4 are not assigned to formulae and therefore not

included in Figure 3.



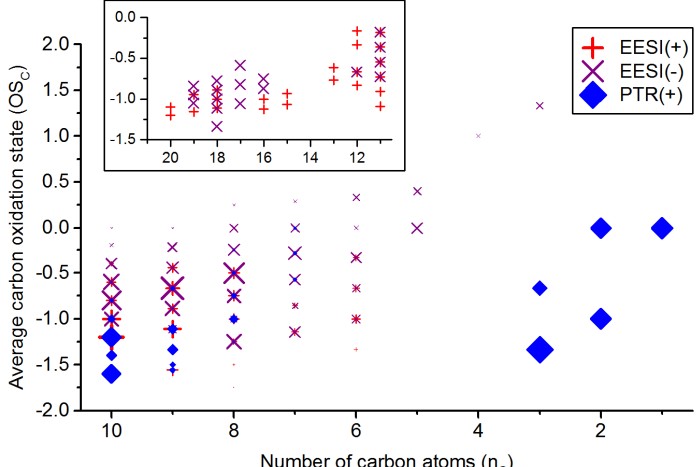

**Figure 3: The average carbon oxidation state of compounds detected via EESI-MS and PTR-MS as a function of $n_C$. Spectra were acquired ~1 hour after the start of ozone introduction for the "medium" concentration experiments. Marker size is proportional to the logarithm of the MS intensity. The inset plot shows oligomeric species ($n_C > 10$) with a constant marker size for clarity. No PTR-MS signals with $n_C > 10$ were detected.**

The ensemble average $\overline{OS_c}$ for particle-phase products resulting from monoterpene ozonolysis is quoted as –1.1 to –0.5 in Kroll et al. (2011) based on ESI-MS and Aerosol Mass Spectrometer (AMS) literature, in good agreement with our EESI(+)/(–)-MS observations. Within this ensemble average, however, the ions detected in the two ion polarities, and in the gas phase via PTR-MS, occupy different regions of the $\overline{OS_c}$-$n_C$ space in Figure 3.

Aside from α-pinene, the products detected by PTR-MS with molecular formula assignments mainly fall into two categories: $n_C$ = 1-3 species resulting from reactions which fragment the α-pinene carbon backbone, and $n_C$ = 8-10 species such as pinonaldehyde resulting from functionalisation reactions. These larger species are sufficiently volatile that a fraction remains in the gas phase, which is reflected in their low average oxidation states ($\overline{OS_c}$ < –0.5). However, many are condensable and therefore also detected in the particle phase via EESI(+), which is generally more sensitive to compounds with only carbonyl groups compared to EESI(–). EESI(+) ions mostly occupy a region with $-1.5 < \overline{OS_c} < -0.5,$ with the most abundant ions assigned as semi-volatile carbonyls such as pinonaldehyde. Carboxylic acids and other functional groups are also present. In addition to





functionalisation products, oligomers from accretion reactions, and products resulting from radical-induced carbon backbone fragmentation, can also be detected.

Compounds detected via EESI(–)-MS were on average more oxidised than EESI(+)-MS and PTR-
MS, and also possess a slightly higher average molecular weight (Figure S4). The negative ion mode is highly efficient at ionising carboxylic acid functional groups, and these heavier and more oxygenated species tend to condense most readily into the particle phase. A number of low $n_C$ species and oligomers were also detected, again indicating that multiple generations of chemistry, including secondary OH-mediated fragmentation, was being assessed. We quantitatively compare EESI(–)-
MS time series for three of the main condensable oxidation products (pinonic acid, pinic acid and OH-pinonic acid) with model simulations in section 3.3.2.

### 3.2.2 Comparison of online EESI(–) and offline nanoESI(–) mass spectra

Figure 4 shows a comparison in the same $\overline{OS_c}$-$n_C$ space as Figure 3 between online measurements
of particle composition via EESI(–)-MS and offline nanoESI(–)-MS analysis following the collection of SOA particles on to quartz fibre filters.

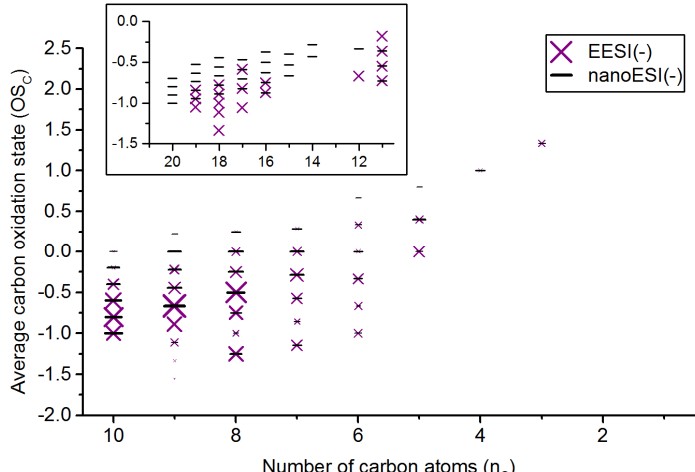



**Figure 4: A comparison of α-pinene SOA oxidation products detected using online EESI(–) and offline nanoESI(–) MS. The EESI(–)-MS spectrum acquisition and offline sampling period started ~1 hour after the start of ozone introduction, for the "medium" concentration experiments. Marker size is proportional to the logarithm of the MS intensity. The inset plot shows oligomeric species ($n_C$ > 10) with a constant marker size for clarity.**

There is good agreement between the two data sets in Figure 4 in terms of the range of $\overline{OS_c}$ and $n_C$ detected, and the relative intensities of species in the main plot (indicated by marker size). Pinic acid ($n_C$ = 9, $\overline{OS_c}$ = –0.67) is the most abundant ion in EESI(–) and nanoESI(–), and essentially all monomeric compounds present with $\overline{OS_c}$ < 0 are detected using both methodologies. A direct comparison of the mass spectra is provided in Figure S5. The major dimers reported in previous studies (Hall IV and Johnston, 2012; Kristensen et al., 2013; Reinhardt et al., 2007) are also detected using both approaches here. Differences between EESI(–) and nanoESI(–) can be observed in two regions: firstly, a few oxidised monomers with $\overline{OS_c}$ > 0 are detected only using the filter method. Secondly, the oligomers detected via nanoESI(–)-MS are more numerous, and generally more oxidised, than for EESI(–)-MS.

The peaks absent from EESI(–)-MS are generally the least abundant in the offline sampling method (Figure S5), so we suspect differences are mainly a consequence of sample pre-concentration and hence improved sensitivity using the filter sampling method. While the EESI source operates at an aerosol flow rate of ~1 L min$^{-1}$, 450 L of chamber air is drawn through the filter for an offline sample. The capture efficiency of aerosols in the EESI process is also expected to be less efficient than extraction from a substrate. An alternative possibility is that offline collection and analysis introduces positive artefacts to the nanoESI(–)-MS spectrum, such as via additional reaction on the substrate. However, the overall good agreement between the methods is encouraging and the use of both during the same experiment could provide the benefits of offline sampling (sensitivity) and online sampling (time resolution, relative quantification) together.

### 3.3 Temporal evolution of molecular composition during α-pinene oxidation

### 3.3.1 Time series of individual particle-phase ions

We now focus on the ability of our online MS techniques to monitor relative concentration changes of individual species during α-pinene oxidation. In particular, we monitor some of the major low volatility products in the particle phase using EESI(–)-MS. Figure 5 shows an illustrative raw time





series from the "medium" experiment conditions for $m/z$ 185.0819, assigned to [M – H]$^-$ for an abundant oxidation products, pinic acid (Table 2). We present the MS data in terms of ion intensities with alternating blank and sample measurements. The pinic acid intensity increases over the first hour of the experiment, as does the total aerosol mass (secondary y-axis in Figure 5). Both signals

tend towards a plateau at later times. Ions assigned to oxidation products in Table 2 show similar increases over time, although the precise time dependence varies depending on the product. We show blank-subtracted time series for other aerosol-phase products in the next section.

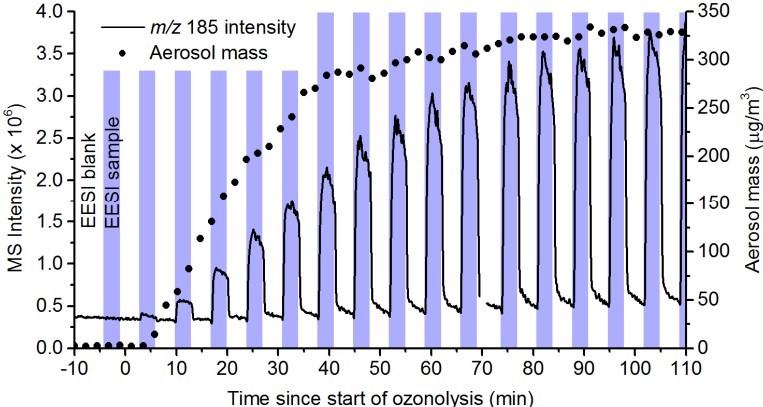

**Figure 5: Left y-axis: Intensity of $m/z$ 185.0819, corresponding to pinic acid, after the start of an α-pinene**
**ozonolysis experiment. Right y-axis: Mass concentration of aerosol in the chamber. The measurements**
**were made using the new EESI-MS batch sampling system (Figure 1). Blue bands correspond to**
**sampling of chamber air from the reservoir volume (3.5 minutes) while the blanks in between correspond**
**to clean air flushing (3.5 minutes). The MS discontinuity around 70 minutes corresponds to refilling of**
**the primary EESI solvent syringe.**

As configured, the sampling setup enables a blank and chamber measurement to be obtained in a seven minute cycle, a substantially higher time resolution than most other semi-continuous sampling methods (Bateman et al., 2009; Pereira et al., 2014) or collection onto filters. It is comparable to the

recent highly time-resolved particle-into-liquid-sampling measurements of Zhang et al., (2015). The signal intensities vary by less than 15 % across a sample window (3.5 minutes) (Figure 5), with most of the decrease attributed to particle deposition in the intermediate reservoir volume and sampling lines. An advantage of flushing the EESI source during each blank period is that baseline changes can be monitored and accounted for. The signals in Figure 5 rapidly return towards the baseline

recorded at the start of the experiment as the source is flushed. Small increases in this baseline (e.g.





due to particle deposition in the source) are subtracted during data processing. Importantly, we avoid the major EESI source contamination problems reported in other applications (McCullough et al., 2011). We suspect this a combination of using relatively low analyte concentrations, the optimised source parameters from Gallimore and Kalberer (2013) and this regular flushing procedure.

### 3.3.2 Comparing measured and modelled time dependence of individual species

An important application of simulation chamber experiments is to better constrain and validate atmospheric reaction mechanisms, particularly for complex VOC chemistry. We compare here individual species measured during the chamber experiments using PTR-MS and EESI-MS to

10 predictions from the AtChem chamber box model using the near-explicit α-pinene oxidation scheme from the Master Chemical Mechanism (referred to as "MCM simulations" from this point). We focus on the first hour of each experiment to assess how the techniques capture the time evolution of composition.

15 We first benchmark the simulations to measurements of [α-pinene] using PTR-MS, an established technique. Figure 6 shows the simulated and measured α-pinene concentrations in the chamber for experiments with [α-pinene]$_0$ = 45, 100 and 500 ppb (referred to as low, medium and high respectively) and [O$_3$]$_{max}$/[α-pinene]$_0$ = 3. Also shown on the secondary y-axis are the corresponding measured SOA concentrations.

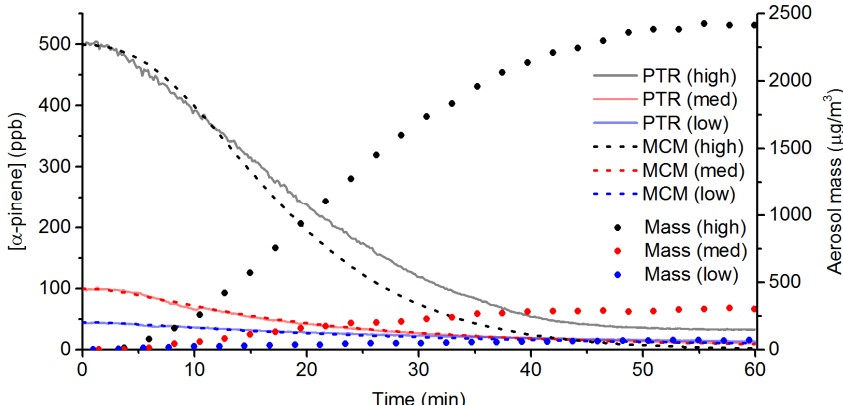

**Figure 6: α-pinene loss and SOA production during chamber experiments with varying [α-pinene]$_0$. Dashed lines: measured [α-pinene] from PTR-MS; Solid lines: MCM simulated [α-pinene]; Points: measured aerosol mass from SMPS.**





The "sigmoidal" shape of the α-pinene loss is a consequence of ozone being introduced over a finite period of up to 30 minutes at the start of the experiment, as discussed in section 2.2. The MCM generally performs well in simulating the observed α-pinene concentration, especially for the low

5 and medium concentration conditions when the ozone introduction period was only a few minutes. An expanded view of the low and medium experiments is provided in Figure S6. The largest discrepancy is for the high concentration experiment, where the simulated loss is more rapid than measured. We attribute this to the longer ozone introduction time in this experiment (~ 30 minutes) which will lead to a more uncertain chamber mixing state at the start of the experiment.

We now demonstrate a comparison of aerosol-phase EESI(–)-MS intensities to the MCM for individual species across the low, medium and high concentration conditions from Table 1. Figure 7 shows time series for *m/z* 183.1027, *m/z* 185.0819 and *m/z* 199.0976, assigned as pinonic acid, pinic acid and OH-pinonic acid respectively (Table 2). The secondary y-axes show simulated

15 concentrations of these species from the MCM under the corresponding conditions.

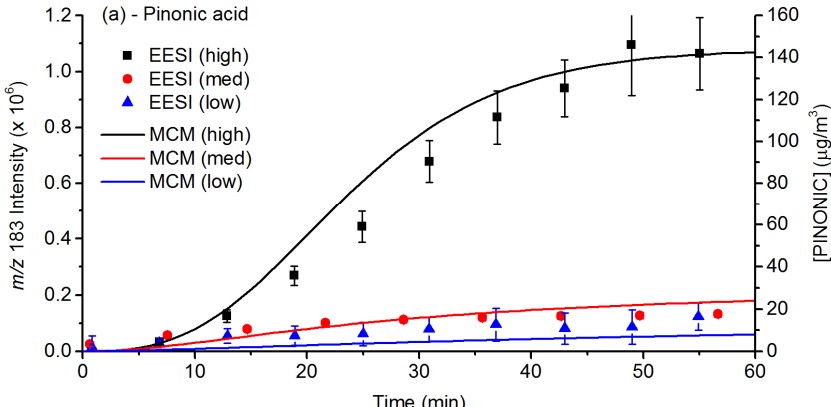




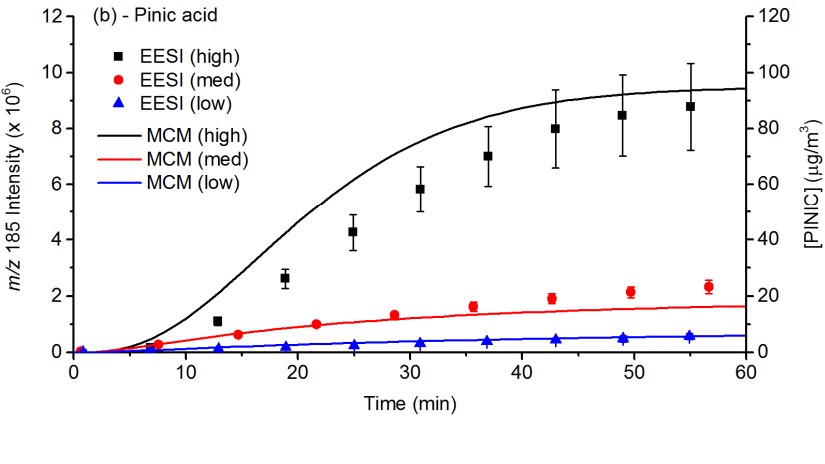

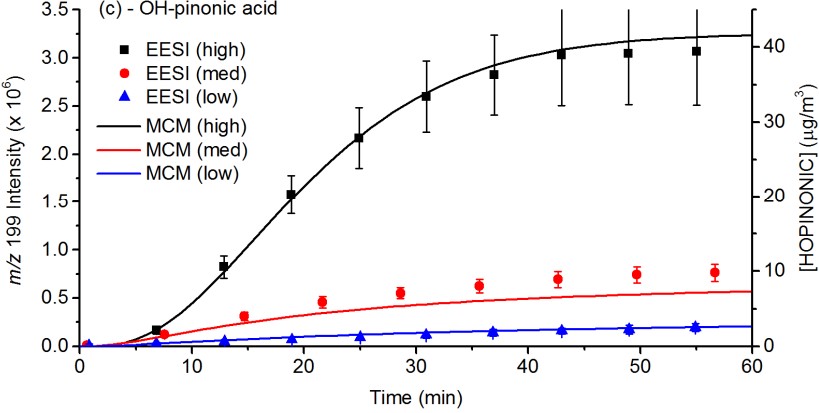

**Figure 7: Comparison between EESI-MS intensities (left y-axis) and MCM simulated concentrations**
5 **(right y-axis) for (a) pinonic acid, (b) pinic acid and (c) OH-pinonic acid during chamber experiments**
**with different [α-pinene]$_0$. MS intensities are absolute, following averaging and blank subtraction. The**
**measurements and model agree closely both in terms of the time dependence of product formation and**
**the relative yields under different reaction conditions.**

10     The overall agreement between MS intensities and MCM simulations is very encouraging. The
measurements and model compare well in two respects: the time dependence of product formation,
and the relative concentrations of a given product in the low, medium and high conditions. Note that



the MS intensities of the three compounds cannot be directly compared without calibration due to the species' different ionisation efficiencies (IEs).

The product time series reflect the rate of consumption of α-pinene in the chamber. As for α-pinene, a pronounced "sigmoidal" profile is observed and simulated for the high concentration conditions, and the slightly overpredicted α-pinene loss rate may explain the more rapid growth of pinonic acid and pinic acid in the model than measured. OH-pinonic acid is a close fit to the model under the range of conditions tested.

The individual product yields after 1 hour (when the α-pinene precursor has been almost completely consumed and timing uncertainties are less important) compare well between measurement and model and scale approximately with $[\alpha\text{-pinene}]_0$. Typically, these relative measured and modelled yields are self-consistent for a given product to within ~25 %, except for pinonic acid in the low concentration experiment. The data for low and medium concentration time series are shown in more detail in Figure S7.

Although pinonic acid, pinic acid and OH-pinonic acid are reported as major particle-phase oxidation products in a range of studies (Table 2), the discussion above assumes that the gas-phase concentrations from the MCM are a good proxy for particle-phase concentrations assessed by EESI-MS. Particle-phase reactions are unlikely to be a large source or sink of these major products, although they may be important for a range of high molecular weight species (Camredon et al., 2010). The equilibrium partitioning of products between the gas and particle phases will favour the particle phase for pinic acid and OH-pinonic acid because their saturation concentrations (~5-10 μg m$^{-3}$, (Müller et al., 2012; Yatavelli et al., 2014)) are substantially lower than aerosol mass loadings (100-2400 μg m$^{-3}$) in our experiments.

Pinonic acid is thought to have a significantly higher saturation concentration (in the range $10^2$-$10^3$ μg m$^{-3}$ depending on the temperature and estimation method used (Müller et al., 2012; Yatavelli et al., 2014)). Hence, it is expected to reside across both phases in our experiments. However, a few observations suggest that the aerosol phase is still favoured here. Gas phase concentrations estimated by PTR-MS are a small fraction of the corresponding MCM predictions in Figure 7(a). The measured EESI-MS intensities also track the predicted MCM concentrations relatively well over time, even as the aerosol mass loading in the chamber is increasing substantially. Zhang et al., (2015) found that aerosol-phase concentrations of pinonic acid increased substantially at 50 % RH (similar to 60% here) compared to dry conditions, and a recent field study concluded that measured pinonic acid





concentrations in the aerosol exceeded absorptive partitioning predictions by a factor of ~ 20 (Yatavelli et al., 2014). Given the relatively high predicted Henry's law coefficient of pinonic acid (~$2 \times 10^7$ M atm$^{-1}$, (Lignell et al., 2013)), it may be that the presence of aerosol liquid water enhances uptake into the aerosol phase. As for all products in our study, structural isomers of the assignments

in Table 2 may also be present as isobaric ions in our mass spectra.

Despite the potential limitations of this comparison, Figure 7 provides further evidence that EESI-MS can be used for relative quantification of individual species in organic aerosols. Moreover, it extends this applicability to scenarios where the particles contain a complex mixture of components,

and where the particle composition, size and total mass are evolving. Specifically, we have demonstrated here that the influence of the bulk aerosol "matrix" on EESI ionisation appears to be negligible up to SOA loadings of ~2400 μg m$^{-3}$ and that the EESI mechanism can tolerate small quantities of inorganic material. The time evolution is also well captured for much lower analyte concentrations of a few μg m$^{-3}$ (Figure S8(c)). This is a significant advantage compared to

conventional direct infusion ESI-MS, where ion intensities are typically only used as a qualitative indicator, if at all, due to strong matrix effects.

Figure 8 shows the measured EESI(–)-MS signal plotted against the corresponding simulated concentration at that time for OH-pinonic acid, which shows the best correlation in time between

measurement and model of the three compounds discussed in Figure 7. This relationship is linear over orders of magnitude in concentration, and regions where the concentrations coincide (e.g. the initial point of the "high" experiment with later points in the "low" and "medium" experiments) overlap well. Figure S8 shows an illustrative plot for all three species from Figure 7, using only the concentrations around 1 hour to minimise timing uncertainties discussed above.





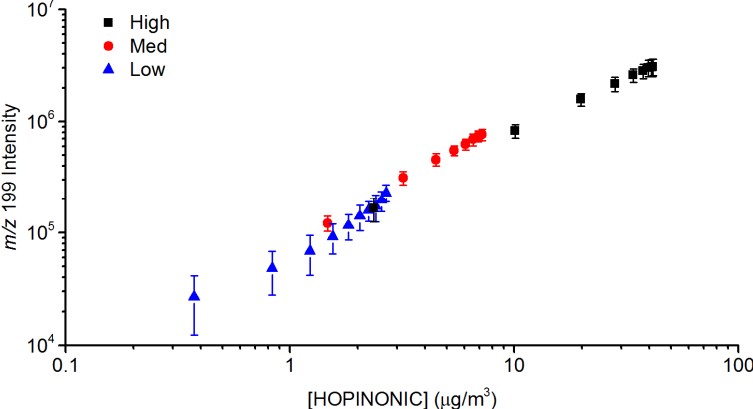

**Figure 8: Correlation between MS intensity and MCM mass across three different experiments for OH-pinonic acid.**

This representation is analogous to the plots for tartaric acid shown in Figure 2, except that the MS signal is compared to the model rather than a known analyte concentration. In principle, a direct calibration curve such as Figure 2 would allow MS intensities to be converted to absolute concentrations for any species where authentic standards are available. In practice, however, the number of species present in aerosols, and the general unavailability of suitable standards, makes

this approach impractical for routine quantification. Comparison to modelled concentrations in this way may therefore provide an approximate indicator of ionisation efficiency for a range of different species in aerosols. However, as discussed above, care is required in interpreting aerosol-phase concentration changes for the many species where in-particle reaction or gas-particle partitioning may be significant. A coupled model of aerosol and gas phase chemistry (Gallimore et al., 2017a;

Shiraiwa et al., 2010) would therefore be a desirable tool to use alongside future chamber experiments.

### 4. Conclusions

Online measurements of particle and gas phase chemistry in the new Cambridge Atmospheric Simulation Chamber (CASC) have been achieved simultaneously using complementary "soft"

ionisation mass spectrometry techniques: Extractive Electrospray Ionisation (EESI) and Proton Transfer Reaction (PTR). The results for EESI-MS are encouraging and prompt continuing use and



further development of the technique in future. Its principle advantages over conventional electrospray techniques are (1) the sample and blank measurements are obtained online, providing highly time resolved information with fewer potential artefacts, and (2) that the ion intensities can be used as a relative measure of concentration due to the lack of matrix interference, and in principle
converted to an absolute concentration via calibration.

The lack of matrix interference in EESI-MS compared to direct ESI-MS has been noted in other applications. The mechanistic differences are not fully understood, but a likely rationale is that the primary electrospray conditions are constant during EESI, but vary substantially depending on the
dissolved analytes in direct ESI. Our work (here and in previous studies) shows a correlation between MS signal and total analyte mass which does not appear to saturate at the upper end of the concentration range tested (~2400 $\mu g\ m^{-3}$). This implies there is an excess of primary charged droplets available to extract and ionise the aerosol components, and hence that the MS intensity of each analyte depends on its relative ionisation efficiency, which has been modelled as a function of
a species' ionisation in solution and ability to delocalise charge (Kruve et al., 2014). Further insight into the EESI mechanism may be obtained by assessing when and how MS saturation behaviour occurs at higher mass loadings.

Improving EESI-MS sensitivity would be an advantage in future atmospheric chemistry
applications. The downside of our Orbitrap mass spectrometer is that the instrument's ion collection and transmission properties are less efficienct relative to other instruments. Coupling the EESI ion source to an alternative mass analyser is an area of active investigation. Pre-concentrating the airborne particles using a virtual impactor system similar to Vogel et al., (2013) may also provide an order-of-magnitude boost to sensitivity.
Although out of the scope of the current study, our molecular composition measurements from the chamber may be amenable to detailed process modelling. A model which includes descriptions of gas-particle partitioning, alongside reactions in both phases, may be better able to capture the dynamic evolution of particle phase components and probe multiphase processing and extended
aging of the initial products.





**Supplementary information**

The supplement contains details related to operation of the atmospheric chamber and EESI sampling system, full EESI-MS, PTR-MS and offline nanoESI-MS spectra, and additional figures comparing the online MS measurements to MCM simulations.

**Acknowledgements**

This work was funded by the European Research Council (grant 279405), the UK Natural Environment Research Council (grant NE/H52449X/1) and the Velux foundation (project number 593).

**Data access**

Data presented in this study can be obtained by contacting the corresponding author.

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
