# Peer review of "Online molecular characterisation of organic aerosols in an atmospheric chamber using Extractive Electrospray Ionisation Mass Spectrometry"

_Atmospheric Chemistry and Physics, 2017_

## Referee Comment (RC1) · Anonymous Referee #1 · 14 Aug 2017

This is an interesting manuscript describing the efficacy of extractive electrospray ionization mass spectrometry (EESI-MS) as an online measurement technique for atmospheric chamber studies. They have used multiple online (EESI-MS and PTR-MS) and offline (ESI-MS) techniques to study the oxidation of $\alpha$-pinene in the CASC chamber. They have shown that EESI-MS can be used to study SOA formed during chamber studies in almost real time (7-minute time resolution). The fact that EESI-MS is a soft ionization technique, it makes it more advantageous over the regular offline ESI-MS studies for chamber studies. They also compared MCM model results to EESI-MS

and PTR-MS results, which showed very good correlations; it shows the potential of EESI-MS as an online measurement tool for chamber studies.

General Comments - Introduction lacks the details of other online measurement techniques (like AMS, and DART-MS) used for $\alpha$-pinene oxidation products, their findings and its comparison to EESI-MS.

- More details about MCM model should be given in the SI in terms of reactions involved in the mechanism. So other laboratory trying to duplicate the work can do so easily.

- All the figures on Oxidation states of carbon against carbon number must include legends what size of the marker represents what intensity. They also should add the correct marker sizes in the inset of each figure, and if that is difficult, it should be included in SI then.

- EESI-MS studies of organic-inorganic particles: why was the study not performed on a compound that is more relevant to this study, for example, pinic acid?

- Although acids have higher sensitivity in negative ion mode, the inorganic salt has NH4+ ions in them, thus positive ion mode would show [M+H]+ and [M+NH4]+ ions for tartaric acid. Assuming all other parameters constant, the ratio of [M+H]+ to [M+NH4]+ should match the mole fraction of the tartaric acid and ammonium sulfate in the atomized solvent. This would be much better way of quantitation than just looking at the acid signal in negative ion mode. I would strongly suggest repeating this particular experiment in positive ion mode.

Specific comment: Page 1, line 30 Low visibility can also be added as an impact.

Page 2 Line 13, The line is talking about soft-ionization in general, however, the reference (Zahardis et al) is not appropriate here. The reference does talk about soft ionization techniques, but the paper is a review of soft ionization techniques for AMS instrument specifically. AMS is mostly used for online studies and not off line measurements. Since the sentence for this reference is used for very generic off-line soft

ionization MS techniques, the reference is not valid here.

Page 2 Line 17 Once again the paragraph is talking about general ESI-MS utility in atmospheric studies. The time resolution for ESI-MS is not hours, not for all kind of studies. However, it will be hours for chamber studies, if authors are refereeing to retention times of hours with respect to chamber studies it should be specified.

Page 3 line 1 Gallimore reference should be 2017a Also, this particular reference is not published yet, thus the conclusion of this study cannot be used as the basis of claiming that EESI-MS can be used for relative quantitation.

Page 4 Line 22 Authors can probably use particle size distributions of Tartaric acid (TA) and Ammonium Sulfate particles to confirm that they are truly internally mixed. If these particles are not internally mixed they probably will show a bimodal distribution. It should be checked for all mole fractions studied, it is possible that higher concentration of ammonium sulfate will lead to a bimodal distribution.

Page 6 Line 10 Was the ESI solvent prepared 1:1 by volume or by weight. Solvent flow rate should also be provided.

Page 6 line 13-14 Authors just described ESI solvent and voltage details previous to these lines. Thus it is really ESI description, not an EESI-MS description. I would suggest adding line 16-19 that describes the particle flow rate to solvent spray before line 13-14.

Figure 1 Should show the position of three-way valve. Page 7 line 6-8 These lines are confusing. Do the authors mean the following? "Air was then pushed through the reservoir and into the EESI source at 1 L min-1 and a sample spectrum was acquired." If yes, it should be corrected. If No, more clarification of the set-up is required. Page 7 line 23 Was formic acid added to the final extract before nano-ESI? If so, details should be added.

Page 7 line 25 Is 1.4kV potential difference correct? Because for EESI it is 3kV. Why

the difference for the same solvent configuration.

Page 9 line 1 Reference and explanation should be provided for the rate constant assumption.

Page 9 line 18 References should be provided for which previous studies were used to identify the major aerosol composition.

Page 10 figure 1 Y axis should be adjusted to show the complete error bar of the first purple triangle marker at a mole fraction of 0.5. Why that particular point has so much error should be explained in the text as well.

Table 2 Once again why wasn't [M+NH4]+ seen in positive ion mode?

Page 13 line 1 The presence of [M+H-H2O]+ peak in Doezema et al studies can be explained in many ways, different instrument parameters for the two studies, different inlet, and mass analyzer voltages etc. Higher capillary Voltage used in that study could simply be due to the combination of solvent composition, solvent flow rate and capillary diameter required to get a good Taylor cone. Authors can only compare the capillary voltage if all other parameters are equal. Page 16 I would suggest moving figure S5 to the main text.

Page 16 line 14-16 If more oxidized species are present in the particle than it could possibly suggest that EESI-MS solvent is extracting some surface of the particle only rather than the entire particle as assumed in Gallimore et al 2013 and I am assuming for this study too. Many studies on oxidation of monoterpenes (Zhao et al 2016 ACP, 16, 3245; Davis et al 2015, Chem.Sci. 6, 7020; Zhao et al 2017, AMT, 10, 1373 ; Trostl et al Nature 2016, 533, 527–531) have suggested that low molecular weight compounds with high Oxidation state are present in the core of the particle and are responsible for new particle formation. Whereas, low molecular weight compounds with low Oxidation state are responsible for the growth of the particle and are usually present on the surface of the particle. Thus the lack of peaks at high m/z in EESI-MS (Figure S5)

suggests that the solvent is extracting the surface of the particle only or at least not the entire particle. Collected particles on the filter analyzed by ESI-MS show many peaks at higher m/z, suggesting ESI is able to sample entire particle, which is expected. But, if the presence of peaks at higher m/z in ESI-MS is due to oligomers (eg 2M+H of pinonaldehyde) than it should be confirmed using MS/MS studies. Identifying the source of these higher m/z peaks in ESI-MS is important to identifying and understand EESI-MS extraction efficiency.

---

## Referee Comment (RC2) · Anonymous Referee #2 · 21 Aug 2017

This manuscript (acp-2017-656) reports the application of extractive electrospray ionisation mass spectrometry (EESI-MS) to the analysis of chamber aerosol produced by reaction of the monoterpene $\alpha$-pinene with O3. Gas phase reactants and products were monitored with proton transfer reaction mass spectrometry (PTR-MS), whereas particle phase products were monitored with EESI-MS with time resolution on the order of 7 min. The time dependence of the MS signals for $\alpha$-pinene (by PTR-MS) and pinonic acid, pinic acid, and OH-pinonic acid (by EESI-MS) correlated well with predictions from the Master Chemical Mechanism (MCM). The signal intensity of reaction

products correlated approximately linearly in log-log space with MCM predictions. Additionally, EESI-MS of test particles containing tartaric acid and ammonium sulfate in known relative molar ratios resulted in a linear signal with increasing mass concentration across all organic-inorganic molar ratios down to 1:1.

This manuscript is well written and the experiments appear to be carefully performed. The figures are well constructed and easy to interpret. This manuscript is within the scope of Atmosphehric Chemistry and Physics and will be suitable for publication once the comments below are satisfactorily addressed.

Comments:

1. One area where this manuscript could be significantly strengthened is in clarifying how much of the EESI-MS signal may arise from gas phase molecules. The chamber contains both organic particles and semivolatile compounds that are going to be present in both the gas phase and the particle phase. It appears the authors use a charcoal denuder to remove the gas phase before sampling into the EESI-MS. However, the authors provide no figure or discussion that quantifies how much, if any, of the EESI-MS signal may arise from gas phase products. The authors should discuss this in the revised manuscript. Resolving whether signal arises from gas or particle could be easily tested by placing a filter to remove the particles before the EESI-MS inlet and monitoring the resulting signal. Note that even if the charcoal denuder removes much of the gas phase, it is possible that compounds that were in the particle phase may partition back to the gas phase during the residence time in the 10 L reservoir. This is an important point, as the authors are comparing their EESI-MS results, which are assumed to be entirely particle phase, to MCM predictions, which is a gas phase model.

2. When the authors examine relative quantification in the mixed organic-inorganic system, they observe that down to a 1:1 organic:inorganic mole fraction the MS signal intensity scales linearly with organic aerosol mass concentration (Fig. 2). The authors

also suggest (page 10, line 20) that suppression of signal may be occurring, but that it is small. It is unclear why the authors did not continue this experiment to higher inorganic aerosol (lower organic aerosol) mole fractions to confirm this suggestion. More discussion is required on this point.

3. The authors should include in the revised manuscript the values for the low mass cut-offs on their mass analysers. As illustrated in Fig. S4 and visualised in Fig. 3, the EESI-MS and PTR-MS spectra appear very different. However, it is not entirely clear if this is due wholly to the different compositions in the gas and particle phase or due to the mass ranges that can be studied with each instrument. The PTR-MS spectrum contains a large number of ions below 100 m/z whereas the EESI-MS spectra do not, but it is not clear in the discussion whether the Orbitrap would necessarily be able to analyse effectively below 100 m/z.

---

## Author Comment (AC1) · 18 Oct 2017

**Responses to Reviewer #1** – "Online molecular characterisation of organic aerosols in an atmospheric chamber using Extractive Electrospray Ionisation Mass Spectrometry" by P.J. Gallimore et al.

Reviewer comments are in blue text, author responses are in black text.

*This is an interesting manuscript describing the efficacy of extractive electrospray ionization mass spectrometry (EESI-MS) as an online measurement technique for atmospheric chamber studies. They have used multiple online (EESI-MS and PTR-MS) and offline (ESI-MS) techniques to study the oxidation of α-pinene in the CASC chamber. They have shown that EESI-MS can be used to study SOA formed during chamber studies in almost real time (7-minute time resolution). The fact that EESI-MS is a soft ionization technique, it makes it more advantageous over the regular offline ESI-MS studies for chamber studies. They also compared MCM model results to EESI-MS and PTR-MS results, which showed very good correlations; it shows the potential of EESI-MS as an online measurement tool for chamber studies.*

We thank the reviewer for the positive appraisal of the manuscript and respond point-by-point to the comments below.

*General Comments - Introduction lacks the details of other online measurement techniques (like AMS, and DART-MS) used for α-pinene oxidation products, their findings and its comparison to EESI-MS.*

We have added some references which use AMS for alpha-pinene SOA (page 2 lines 15-17): "A great deal of insight into SOA formation and aging from monoterpenes has been provided by established instruments such as the Aerosol Mass Spectrometer (Aiken et al., 2008; Chhabra et al., 2010; Shilling et al., 2009)."

We could not find specific uses of DART-MS for alpha-pinene SOA but have added references to Nah et al., (2013) and Zhao et al., (2017) as examples of the useful general insight provided by the technique (page 2 lines 30-31).

*- More details about MCM model should be given in the SI in terms of reactions involved in the mechanism. So other laboratory trying to duplicate the work can do so easily.*

We have included an additional section in the Supplement (page 4):

**"Reaction scheme for MCM simulations**

The reaction scheme for the degradation of α-pinene is based on the mechanism described in Saunders et al., (2003). All reactions involving the ozone-initiated oxidation of α-pinene and its products, including decomposition of reaction intermediates and secondary OH-mediated chemistry, were extracted from the MCM website (http://mcm.leeds.ac.uk/MCM) in June 2017. This mechanism extract comprises 313 explicit species.

We also included an auxiliary reaction to approximate the gradual introduction of ozone into the chamber (reaction S2):

$$\text{pre-O}_3 \text{ (g)} \rightarrow \text{O}_3 \text{ (g)} \qquad k_{pre} \qquad\qquad\qquad \text{(S2)}$$

Pre-O$_3$ does not participate in any other reactions, and is converted into O$_3$ on a timescale $\tau_{pre} = 1/k_{pre}$ = 3-30 minutes, adjusted to reflect the experimental timescale for ozone introduction for each experiment.

The simulations were initialised with all species concentrations set to zero, apart from: α-pinene (MCM name APINENE), which was set to 45, 100 and 500 ppb for the low, medium and high concentration experiments respectively, and pre-O$_3$ (PREO3) which was adjusted to simulate the observed [O$_3$]$_{max}$ in each experiment (Table 1)."

*- All the figures on Oxidation states of carbon against carbon number must include legends what size of the marker represents what intensity. They also should add the correct marker sizes in the inset of each figure, and if that is difficult, it should be included in SI then.*

We have adapted Figures 3 and 4 to include a scale relating intensity to marker size, using a binned intensity scale for simplicity. This now also includes the inset regions. The figure captions have been updated accordingly (page 15 lines 3-5 and page 21 lines 1-3).

*- EESI-MS studies of organic-inorganic particles: why was the study not performed on a compound that is more relevant to this study, for example, pinic acid?*

We have added the following discussion (page 10, lines 25-28): "Tartaric acid is a highly oxygenated compound which is a relevant proxy for species present in aged organic aerosols. Its high water solubility and negligible volatility allow mixed TA-AS particles with precisely known composition to be prepared from solution."

As noted on page 11 line 7, The use of TA also enables direct comparison with previous quantification work performed in Gallimore and Kalberer (2013).

*- Although acids have higher sensitivity in negative ion mode, the inorganic salt has NH4+ ions in them, thus positive ion mode would show [M+H]+ and [M+NH4]+ ions for tartaric acid. Assuming all other parameters constant, the ratio of [M+H]+ to [M+NH4]+ should match the mole fraction of the tartaric acid and ammonium sulfate in the atomized solvent. This would be much better way of quantitation than just looking at the acid signal in negative ion mode. I would strongly suggest repeating this particular experiment in positive ion mode.*

We thank the reviewer for this thoughtful suggestion. However, we feel that quantification in positive ion mode is out of the scope of the current paper given that the subsequent emphasis on quantification uses negative ion mode (online/offline comparison, time series analysis). Instead, it would be better suited to an entire additional study.

It is not clear to us that [M+H]$^+$/[M+NH$_4$]$^+$ would necessarily scale in a simple manner. This will depend on competitive ionisation processes – the relative affinities of the analyte for H$^+$ and NH$_4^+$, competition from contaminant Na$^+$, and the absolute number of ions in the primary spray. We suspect the latter is far in excess of the number of analyte molecules extracted given that the relative quantification presented in Figure 2

and in Gallimore and Kalberer (2013) does not appear to "saturate" at higher mass loadings.

We have added a sentence to emphasise the purpose of section 3.1 (page 10 lines 21-23): "Specifically, we establish the potential impact of inorganic seed particles on the relative quantification of organic acids (detected as $[M–H]^-$) in the chamber experiments which follow."

*Specific comment: Page 1, line 30 Low visibility can also be added as an impact.*

We now mention this as an impact (page 2 line 3).

*Page 2 Line 13, The line is talking about soft-ionization in general, however, the reference (Zahardis et al) is not appropriate here. The reference does talk about soft ionization techniques, but the paper is a review of soft ionization techniques for AMS instrument specifically. AMS is mostly used for online studies and not off line measurements. Since the sentence for this reference is used for very generic off-line soft ionization MS techniques, the reference is not valid here.*

We have replaced the Zahardis et al. reference with Hoffmann et al., (2011) (page 2 line 19), which features a discussion of soft ionisation MS techniques in the wider context of organic aerosol analysis.

*Page 2 Line 17 Once again the paragraph is talking about general ESI-MS utility in atmospheric studies. The time resolution for ESI-MS is not hours, not for all kind of studies. However, it will be hours for chamber studies, if authors are refereeing to retention times of hours with respect to chamber studies it should be specified.*

We have added clarification to our previous comments about time resolution (page 2 lines 23-25): "…the time resolution of measurements depends on the frequency at which aerosol samples are collected, which is typically an hour or more for chamber and ambient sampling."

*Page 3 line 1 Gallimore reference should be 2017a Also, this particular reference is not published yet, thus the conclusion of this study cannot be used as the basis of claiming that EESI-MS can be used for relative quantitation.*

The reference has now been published; the full citation is given in the References section and is referred to as Gallimore et al., (2017a) in the text (page 3, line 10), consistent with the rest of the manuscript.

*Page 4 Line 22 Authors can probably use particle size distributions of Tartaric acid (TA) and Ammonium Sulfate particles to confirm that they are truly internally mixed. If these particles are not internally mixed they probably will show a bimodal distribution. It should be checked for all mole fractions studied, it is possible that higher concentration of ammonium sulfate will lead to a bimodal distribution.*

Thank you for this insight. We have added (page 4, lines 27-28): "A single mode was observed in the particle size distribution for all nebuliser solutions; particles were therefore assumed to be internally mixed…"

*Page 6 Line 10 Was the ESI solvent prepared 1:1 by volume or by weight. Solvent flow rate should also be provided.*

We now mention (page 6, line 21): "…a water-methanol 1:1 mixture by volume…" and state (page 6, lines 23-24): "The solvent flow rate was set to 10 µL min$^{-1}$."

*Page 6 line 13-14 Authors just described ESI solvent and voltage details previous to these lines. Thus it is really ESI description, not an EESI-MS description. I would suggest adding line 16-19 that describes the particle flow rate to solvent spray before line 13-14.*

We have moved the particle flow discussion above the first mention of EESI(+) and EESI(−) operating conditions (page 6, line 26).

*Figure 1 Should show the position of three-way valve.*

Figure 1 has been updated accordingly (page 7), and also now includes the optional HEPA filter discussed in response to Reviewer 2.

*Page 7 line 6-8 These lines are confusing. Do the authors mean the following? "Air was then pushed through the reservoir and into the EESI source at 1 L min-1 and a sample spectrum was acquired." If yes, it should be corrected. If No, more clarification of the set-up is required.*

We have corrected "flow tube" to "reservoir" as suggested (page 7 line 14).

*Page 7 line 23 Was formic acid added to the final extract before nano-ESI? If so, details should be added.*

We now mention (page 8, lines 5-6): "…and 0.1 % by volume formic acid was added to the extract"

*Page 7 line 25 Is 1.4kV potential difference correct? Because for EESI it is 3kV. Why the difference for the same solvent configuration.*

This potential difference is correct. The lower solvent low rates used in nanoESI are commonly paired with lower potential differences compared to conventional ESI. We have not updated the text.

*Page 9 line 1 Reference and explanation should be provided for the rate constant assumption.*

We have added the following sentence (page 9, lines 24-26): "Exothermic proton transfer reactions occur at a rate close to the collision limit (Blake et al., 2009) and known rate constants for a variety of VOCs are within 50 % of this value (Smith and Spanel, 2005)."

*Page 9 line 18 References should be provided for which previous studies were used to identify the major aerosol composition.*

We now refer to "previous studies referenced in Table 2" (page 10 lines 12-13) which provides a summary of major products considered in this manuscript.

*Page 10 figure 1 Y axis should be adjusted to show the complete error bar of the first purple triangle marker at a mole fraction of 0.5. Why that particular point has so much error should be explained in the text as well.*

The y-axis of Figure 2 has now been rescaled accordingly (page 11). The absolute error on this marker is low, but, following background subtraction, appears large because the data are presented on a log scale. We have not changed the text.

*Table 2 Once again why wasn't [M+NH4]+ seen in positive ion mode?*

We have added the following discussion (page 8, lines 20-29): "This removes formulae which fall outside a 2 ppm mass tolerance and those deemed implausible based on their atomic ratios. By strictly limiting permitted elements, we reduce the number of erroneous permutations of formulae that coincide with the measured $m/z$. Based on the oxygenated VOCs expected for α-pinene SOA (Table 2), we retained assignments containing only carbon, hydrogen and oxygen, and permitted sodium adduct formation in positive ion mode. Nitrogen-containing compounds are not expected to form via ozonolysis, but we note that this treatment excludes possible $[M + NH_4]^+$ adducts derived from the ammonium sulfate seed particles. However, in the raw data, such clusters were not apparent for major positive mode species such as pinonaldehyde."

As discussed above in reply to general comment #5, we speculate that this is a consequence of the apparent large excess of primary electrospray ions (favouring $H^+$ and $Na^+$) relative to organic and ammonium analytes.

*Page 13 line 1 The presence of [M+H-H2O]+ peak in Doezema et al studies can be explained in many ways, different instrument parameters for the two studies, different inlet, and mass analyzer voltages etc. Higher capillary Voltage used in that study could simply be due to the combination of solvent composition, solvent flow rate and capillary diameter required to get a good Taylor cone. Authors can only compare the capillary voltage if all other parameters are equal.*

We agree that the other parameters mentioned by the reviewer may also be important and have deleted the specific reference to voltage. The revised sentence now reads (page 14, lines 1-3): "This suggests that the choice of EESI parameters may be an important determinant of fragmentation."

*Page 16 I would suggest moving figure S5 to the main text.*

We have included this as Figure 5 in the main text (page 17) and deleted it from the Supplement. Later figures have been renamed accordingly.

*Page 16 line 14-16 If more oxidized species are present in the particle than it could possibly suggest that EESI-MS solvent is extracting some surface of the particle only rather than the entire particle as assumed in Gallimore et al 2013 and I am assuming for this study too. Many studies on oxidation of monoterpenes (Zhao et al 2016 ACP, 16, 3245; Davis et al 2015,*

*Chem.Sci. 6, 7020; Zhao et al 2017, AMT, 10, 1373 ; Trostl et al Nature 2016, 533, 527–531) have suggested that low molecular weight compounds with high Oxidation state are present in the core of the particle and are responsible for new particle formation. Whereas, low molecular weight compounds with low Oxidation state are responsible for the growth of the particle and are usually present on the surface of the particle. Thus the lack of peaks at high m/z in EESI-MS (Figure S5) suggests that the solvent is extracting the surface of the particle only or at least not the entire particle. Collected particles on the filter analyzed by ESI-MS show many peaks at higher m/z, suggesting ESI is able to sample entire particle, which is expected. But, if the presence of peaks at higher m/z in ESI-MS is due to oligomers (eg 2M+H of pinonaldehyde) than it should be confirmed using MS/MS studies. Identifying the source of these higher m/z peaks in ESI-MS is important to identifying and understand EESI-MS extraction efficiency.*

It has been shown that other MS techniques such as Direct Analysis in Real Time (DART) only analyse an outer portion of particles (Nah et al., 2013). However, our tests on EESI-MS, both here (Figure 2) and previously in Gallimore and Kalberer (2013), show the MS signal scales with total particle mass, independent of particle size for the range tested. While the EESI ionisation process is not yet fully understood (Law et al., 2010), we believe aerosol components in the core of particles are at least available for extraction and ionisation under these conditions. Further work is required to establish at what particle size and/or total mass this linearity may break down.

Higher $m/z$ ions were identified using both ESI-MS and EESI-MS analyses here (see Figure 4 and the new Figure 5). Many have been positively identified as in-particle oligomers (rather than ionisation artefacts) in previous studies (e.g. (Kristensen et al., 2013)). As discussed (page 18, lines 1-3), our hypothesis is that the larger number seen in the offline analysis is a concentration/sensitivity effect. We should also point out that offline ESI-MS is known to suffer from positive and negative sampling artefacts, and competitive ionisation effects. It therefore does not necessarily provide a definitive point of comparison for the true "bulk" composition.

We have added an additional sentence (page 18, lines 9-11): "Horan et al., (2012) found that the relative abundance of oligomers was also higher in filter samples than their online AESI method, which was attributed to a negative filter sampling artefact – evaporation of semi-volatile material (predominantly monomers) during collection."

**References**

Aiken, A. C., Decarlo, P. F., Kroll, J. H., Worsnop, D. R., Huffman, J. A., Docherty, K. S., Ulbrich, I. M., Mohr, C., Kimmel, J. R., Sueper, D., Sun, Y., Zhang, Q., Trimborn, A., Northway, M. J., Ziemann, P. J., Canagaratna, M. R., Onasch, T. B., Alfarra, M. R., Prevot, A. S. H., Dommen, J., Duplissy, J., Metzger, A., Baltensperger, U. and Jimenez, J. L.: O / C and OM / OC Ratios of Primary , Secondary , and Ambient Organic Aerosols with High-Resolution Time-of-Flight Aerosol Mass Spectrometry, Environ. Sci. Technol, 42(12), 4478–4485, 2008.

Blake, R. S., Monks, P. S. and Ellis, A. M.: Proton-transfer reaction mass

spectrometry., Chem. Rev., 109(3), 861–96, doi:10.1021/cr800364q, 2009.

Chhabra, P. S., Flagan, R. C. and Seinfeld, J. H.: Elemental analysis of chamber organic aerosol using an aerodyne high-resolution aerosol mass spectrometer, Atmos. Chem. Phys., 10, 4111–4131, doi:10.5194/acp-10-4111-2010, 2010.

Gallimore, P. J. and Kalberer, M.: Characterizing an extractive electrospray ionization (EESI) source for the online mass spectrometry analysis of organic aerosols, Environ. Sci. Technol., 47(13), 7324–31, doi:10.1021/es305199h, 2013.

Gallimore, P. J., Griffiths, P. T., Pope, F. D., Reid, J. P. and Kalberer, M.: Comprehensive modeling study of ozonolysis of oleic acid aerosol based on real-time, online measurements of aerosol composition, J. Geophys. Res. Atmos., 122, 4364–4377, doi:10.1002/2016JD026221, 2017a.

Hoffmann, T., Huang, R.-J. and Kalberer, M.: Atmospheric Analytical Chemistry, Anal. Chem., 83(12), 4649–64, doi:10.1021/ac2010718, 2011.

Horan, A. J., Gao, Y., Hall, W. A. and Johnston, M. V: Online Characterization of Particles and Gases with an Ambient Electrospray Ionization Source, Anal. Chem., 84, 9253–9258, 2012.

Kristensen, K., Enggrob, K. L., King, S. M., Worton, D. R., Platt, S. M., Mortensen, R., Rosenoern, T., Surratt, J. D., Bilde, M., Goldstein, A. H. and Glasius, M.: Formation and occurrence of dimer esters of pinene oxidation products in atmospheric aerosols, Atmos. Chem. Phys., 13(7), 3763–3776, doi:10.5194/acp-13-3763-2013, 2013.

Law, W. S., Wang, R., Hu, B., Berchtold, C., Meier, L., Chen, H. and Zenobi, R.: On the mechanism of extractive electrospray ionization., Anal. Chem., 82(11), 4494–500, doi:10.1021/ac100390t, 2010.

Nah, T., Chan, M., Leone, S. R. and Wilson, K. R.: Real Time in Situ Chemical Characterization of Submicrometer Organic Particles Using Direct Analysis in Real Time-Mass Spectrometry, Anal. Chem., 85, 2087–95, doi:10.1021/ac302560c, 2013.

Saunders, S. M., Jenkin, M. E., Derwent, R. G. and Pilling, M. J.: Protocol for the development of the Master Chemical Mechanism, MCM v3 (Part A): tropospheric degradation of non-aromatic volatile organic compounds, Atmos. Chem. Phys., 3, 161–180, 2003.

Shilling, J. E., Chen, Q., King, S. M., Rosenoern, T., Kroll, J. H., Worsnop, D. R., DeCarlo, P. F., Aiken, a. C., Sueper, D., Jimenez, J. L. and Martin, S. T.: Loading-dependent elemental composition of α-pinene SOA particles, Atmos. Chem. Phys., 9, 771–782, doi:10.5194/acp-9-771-2009, 2009.

Smith, D. and Spanel, P.: SELECTED ION FLOW TUBE MASS SPECTROMETRY ( SIFT-MS ) FOR ON-LINE TRACE GAS ANALYSIS, Mass Spectrom. Rev., 661–700, doi:10.1002/mas.20033, 2005.

Zhao, Y., Fairhurst, M. C., Wingen, L. M., Perraud, V. and Ezell, M. J.: New insights into atmospherically relevant reaction systems using direct analysis in real-time mass spectrometry ( DART-MS ), , 1373–1386, doi:10.5194/amt-10-1373-2017, 2017.

---

## Author Comment (AC2) · 18 Oct 2017

**Responses to Reviewer #2** – "Online molecular characterisation of organic aerosols in an atmospheric chamber using Extractive Electrospray Ionisation Mass Spectrometry" by P.J. Gallimore et al.

Reviewer comments are in blue text, author responses are in black text.

*This manuscript (acp-2017-656) reports the application of extractive electrospray ionisation mass spectrometry (EESI-MS) to the analysis of chamber aerosol produced by reaction of the monoterpene α-pinene with O3. Gas phase reactants and products were monitored with proton transfer reaction mass spectrometry (PTR-MS), whereas particle phase products were monitored with EESI-MS with time resolution on the order of 7 min. The time dependence of the MS signals for α-pinene (by PTR-MS) and pinonic acid, pinic acid, and OH-pinonic acid (by EESI-MS) correlated well with predictions from the Master Chemical Mechanism (MCM). The signal intensity of reaction products correlated approximately linearly in log-log space with MCM predictions. Additionally, EESI-MS of test particles containing tartaric acid and ammonium sulfate in known relative molar ratios resulted in a linear signal with increasing mass concentration across all organic-inorganic molar ratios down to 1:1.*

*This manuscript is well written and the experiments appear to be carefully performed. The figures are well constructed and easy to interpret. This manuscript is within the scope of Atmospehric Chemistry and Physics and will be suitable for publication once the comments below are satisfactorily addressed.*

We thank the reviewer for the positive appraisal of the manuscript and respond to the comments below.

*Comments:*

*1. One area where this manuscript could be significantly strengthened is in clarifying how much of the EESI-MS signal may arise from gas phase molecules. The chamber contains both organic particles and semivolatile compounds that are going to be present in both the gas phase and the particle phase. It appears the authors use a charcoal denuder to remove the gas phase before sampling into the EESI-MS. However, the authors provide no figure or discussion that quantifies how much, if any, of the EESI-MS signal may arise from gas phase products. The authors should discuss this in the revised manuscript. Resolving whether signal arises from gas or particle could be easily tested by placing a filter to remove the particles before the EESI-MS inlet and monitoring the resulting signal. Note that even if the charcoal denuder removes much of the gas phase, it is possible that compounds that were in the particle phase may partition back to the gas phase during the residence time in the 10 L reservoir. This is an important point, as the authors are comparing their EESI-MS results, which are assumed to be entirely particle phase, to MCM predictions, which is a gas phase model.*

We agree that this is an insightful additional test and followed the reviewer's suggestion of performing additional experiments using a filter. The manuscript has been modified in the following places:

Page 7, lines 15-17: "An optional High Efficiency Particle Air (HEPA) filter (HEPA CAP, Whatman) was used to test the possible contribution of gas phase species to the observed MS signal."

Page 19 line 24-page 20 line 15: "The semi-volatile nature of SOA means that both gas- and particle-phase species will be present in the chamber. We examined whether gas-phase species contribute to our observed EESI(−)-MS signal under the "medium" reaction conditions by removing particles from the sample flow using a HEPA filter (Figure 1). With particles filtered out, none of the species listed in Table 2 could be detected above levels observed for the solvent blanks. This was also the case even if the charcoal denuder in Figure 1, intended to remove gas-phase species, was bypassed.

The aerosol mass loading in the chamber ($\sim$300 $\mu g\,m^{-3}$) would strongly bias most of the compounds in Table 2 towards the particle phase. For instance, the most abundant ion, pinic acid, has a vapour pressure $\sim$ 2 $\mu g\,m^{-3}$ at 294 K (Bilde and Pandis, 2001), so > 99% would be expected to reside in the particle phase based on an absorptive partitioning argument (Pankow, 1994). This might explain the lack of detected species in the present SOA system. However, a number of studies have detected gas-phase species using an electrospray source, e.g. (Horan et al., 2012; Wu et al., 2000; Zhao et al., 2017). The ion source design and operating parameters appear important in determining the ionisation efficiency and mechanism (uptake into droplets or gas-phase chemical ionisation). Future work to simultaneously detect semi-volatile species in both phases, and understand the relative efficiencies of gas- and particle-phase ionisation, is therefore merited."

Abstract (page 1, lines 19-20): "Under our experimental conditions, EESI-MS signals arise only from particle-phase analytes."

Conclusions (page 26, line 25-27): "Our limited tests with this EESI-MS configuration show that the signal arising is a result of droplet-particle collisions, with negligible contribution from gas-phase analytes."

*2. When the authors examine relative quantification in the mixed organic-inorganic system, they observe that down to a 1:1 organic:inorganic mole fraction the MS signal intensity scales linearly with organic aerosol mass concentration (Fig. 2). The authors also suggest (page 10, line 20) that suppression of signal may be occurring, but that it is small. It is unclear why the authors did not continue this experiment to higher inorganic aerosol (lower organic aerosol) mole fractions to confirm this suggestion. More discussion is required on this point.*

We have added the following discussion to emphasise the purpose of Figure 2 (page 10, lines 21-23): "Specifically, we establish the potential impact of inorganic seed particles on the relative quantification of organic acids (detected as $[M–H]^-$) in the chamber experiments which follow." 5 $\mu g\,m^{-3}$ ammonium sulfate seed particles were added to the chamber, and this corresponded to small fractions of the aerosol mass during MS analysis. These conditions are covered by Figure 2.

We agree that considering a wider parameter space (including particle composition) in future is desirable to extend the applicability, and understand potential limitations of, the EESI-MS technique.

*3. The authors should include in the revised manuscript the values for the low mass cut-offs on their mass analysers. As illustrated in Fig. S4 and visualised in Fig. 3, the EESI-MS and PTR-MS spectra appear very different. However, it is not entirely clear if this is due wholly to the different compositions in the gas and particle phase or due to the mass ranges that can be studied with each instrument. The PTR-MS spectrum contains a large number of ions below 100 m/z whereas the EESI-MS spectra do not, but it is not clear in the discussion whether the Orbitrap would necessarily be able to analyse effectively below 100 m/z.*

The mass ranges for the mass analysers are already stated in the methods section (page 8, line 12: *m/z* 100-500 for EESI-MS; page 9 line 9: *m/z* 10-500 for PTR-MS). We have added the following emphasis in the discussion (page 12, lines 14-15): "The low mass thresholds of the EESI-MS and PTR-MS mass analysers were *m/z* 100 and 10 respectively in this study."

**References**

Bilde, M. and Pandis, S. N.: Evaporation Rates and Vapor Pressures of Individual Aerosol Species Formed in the Atmospheric Oxidation of alpha- and beta-Pinene, Environ. Sci. Technol., 35(16), 3344–3349, 2001.

Horan, A. J., Gao, Y., Hall, W. A. and Johnston, M. V: Online Characterization of Particles and Gases with an Ambient Electrospray Ionization Source, Anal. Chem., 84, 9253–9258, 2012.

Pankow, J. F.: An absorption model of the gas/aerosol partitioning involved in the formation of secondary organic aerosol, Atmos. Environ., 28(2), 189–193, 1994.

Wu, C., Siems, W. F. and Hill Jr., H. H.: Secondary Electrospray Ionization Ion Mobility Spectrometry / Mass Spectrometry of Illicit Drugs, Anal. Chem., 72(2), 396–403, 2000.

Zhao, Y., Chan, J. K., Lopez-Hilfiker, F. D., McKeown, M. A., Ambro, E. L. D., Slowik, J. G., Riffell, J. A. and Thornton, J. A.: An electrospray chemical ionization source for real-time measurement of atmospheric organic and inorganic vapors, Atmos. Meas. Tech., 3609–3625, 2017.

---

## Author Response (AR2)

[revised manuscript text omitted]

**Reaction scheme for MCM simulations**

The reaction scheme for the degradation of α-pinene is based on the mechanism described in Saunders et al., (2003). All reactions involving the ozone-initiated oxidation of α-pinene and its products, including decomposition of reaction intermediates and secondary OH-mediated chemistry,

5   were extracted from the MCM website (http://mcm.leeds.ac.uk/MCM) in June 2017. This mechanism extract comprises 313 explicit species.

We also included an auxiliary reaction to approximate the gradual introduction of ozone into the chamber (reaction S2):

$$\text{pre-O}_3 \text{ (g)} \rightarrow \text{O}_3 \text{ (g)} \qquad k_{pre} \qquad \qquad \text{(S2)}$$

Pre-O$_3$ does not participate in any other reactions, and is converted into O$_3$ on a timescale $\tau_{pre} = 1/k_{pre}$ = 3-30 minutes, adjusted to reflect the experimental timescale for ozone introduction for each experiment.

15   The simulations were initialised with all species concentrations set to zero, apart from: α-pinene (APINENE), which was set to 45, 100 and 500 ppb for the low, medium and high concentration experiments respectively, and pre-O$_3$ (PREO3) which was adjusted to simulate the observed [O$_3$]$_{max}$ in each experiment (Table 1).

**Extraction efficiency of tartaric acid in mixed organic-inorganic aerosols**

[Figure]

**Figure S3: Tartaric acid MS signal, normalised by the tartaric acid aerosol mass, as a function of the ammonium sulfate aerosol mass. The blue solid and dotted lines represent  the mean and standard deviation respectively of measurements on single-component tartaric acid aerosols ($x_{TA} = 1$).**

**Mass spectra for EESI-MS and PTR-MS**

[Figure]

**Figure S4: Mass spectra obtained during the dark ozonolysis of α-pinene under "medium" conditions (Table 1) using three online MS techniques: (a) EESI-MS in positive ionisation mode, (b) EESI-MS in negative ionisation mode and (c) PTR-MS using $H_3O^+$ as a chemical ionisation reagent. The ions have been assigned to molecular formulae and are plotted as neutral masses to aid comparison. Only PTR-MS ions corresponding to assigned α-pinene ozonolysis products are shown. Taken together, the three techniques enable online detection of a wide variety of organic compound classes and volatilities, from hydrocarbons such as α-pinene to highly oxidised carboxylic acids.**

**Comparison between measured MS peak abundances and MCM concentrations for oxidation products**

[Figure]

**Figure S5: α-pinene loss and SOA production during chamber experiments with varying [α-pinene]$_0$, showing only the medium and low concentration conditions. Dashed lines: measured [α-pinene] from PTR-MS; Solid lines: MCM simulated [α-pinene]; Points: measured aerosol mass from SMPS.**

[Figure]

[Figure]

Figure S6: Comparison between EESI-MS peak abundances (left y-axis) and MCM simulated concentrations (right y-axis) for (a) pinonic acid, (b) pinic acid and (c) OH-pinonic acid, showing only the medium and low concentration conditions. The scales of the y-axes from Figure 8 have been divided by 4, so that the relative scaling between the *m/z* and MCM axes is consistent with Figure 8.

[Figure]

**Figure S7: MS peak abundance of pinonic, pinic and OH-pinonic acids as a function of predicted MCM concentrations. The data shown are for the measurements approximately 1 hour after the start of ozonolysis in Figure 8, where the rate of change of concentration slows and hence simulated and measured concentrations represent approximate final yields.**

We have removed these unnecessary commas.

P3,L9: missing space

A space has been added.

P4,L13: Calculations of mole fractions normally include the number of moles of solvent in the denominator. The mole fractions cited here refer to the solutes only, without solvent. I presume the readers will figure it out but revising the text to remove this ambiguity would help.

We have added a clarification: "The total solute concentration was held constant at 0.1 mol L$^{-1}$. The solute mole fractions (excluding water) were varied…"

P6,L3: not used -> was not used

This has been changed.

P6, L15: is described -> was described

This has been changed.

P11, Figure 2, also applies to other figures: I have been told by several mass spectrometrists that the use of term "intensity" is now discouraged in mass spectrometry literature. Terms like "ion abundance", "relative ion abundance", etc. appear to be preferred. If you adopt this convention, you will need to change labels in your figures and figure captions.

We have adopted the terminology of "abundance" in place of "intensity" throughout the manuscript text, figures and captions.

P14, L4: using "EESI-MS(–)" might be a little less awkward than "EESI(–)-MS"

While both are slightly awkward, we have retained the current notation to remain consistent with reference to "EESI(–)" (without "MS") and related terms in other parts of the manuscript.

P23, L11 and L25: unnecessary commas after the stated values

These have been removed.

P24, Figure 9: I would edit the X-axis label to say "[OH-pinonic acid]" instead of "[OHPINONIC]"

We have changed the label as suggested.

Reference section:

Please fix subscripts/superscripts, for example in Edney et al. (2005)

We have corrected subscripts and superscripts as suggested.

Please fix title capitalization in Smith and Spanel (2005). It would also help to make the capitalization more uniform.

The capitalisation has been removed.

Supporting information section:

P1,L2: apparatus … are shown -> setups … are shown

This has been corrected.

P1, L13: instrumentation is -> instruments are

We have changed this.

P4,L3: Saunders et al., -> Saunders et al.

The comma has been removed.

P5,L4: represent represents -> represent

"Represents" has been deleted.

Figure S4, S5, S6, S7: "peak abundance" may be a more accurate in the Y-axis title (see above)

We have modified these figures accordingly.